# Genes and Microbiota Interaction in Monogenic Autoimmune Disorders

**DOI:** 10.3390/biomedicines11041127

**Published:** 2023-04-08

**Authors:** Federica Costa, Eleonora Beltrami, Simona Mellone, Sara Sacchetti, Elena Boggio, Casimiro Luca Gigliotti, Ian Stoppa, Umberto Dianzani, Roberta Rolla, Mara Giordano

**Affiliations:** 1Department of Health Sciences, Università del Piemonte Orientale, 28100 Novara, Italy; 20042482@studenti.uniupo.it (F.C.); 20016248@studenti.uniupo.it (S.S.); elena.boggio@med.uniupo.it (E.B.); luca.gigliotti@med.uniupo.it (C.L.G.); ian.stoppa@uniupo.it (I.S.); roberta.rolla@med.uniupo.it (R.R.); mara.giordano@med.uniupo.it (M.G.); 2Maggiore della Carità University Hospital, 28100 Novara, Italy; eleonora.beltrami@maggioreosp.novara.it (E.B.); simona.mellone@uniupo.it (S.M.)

**Keywords:** monogenic disease, autoimmunity, microbiota

## Abstract

Monogenic autoimmune disorders represent an important tool to understand the mechanisms behind central and peripheral immune tolerance. Multiple factors, both genetic and environmental, are known to be involved in the alteration of the immune activation/immune tolerance homeostasis typical of these disorders, making it difficult to control the disease. The latest advances in genetic analysis have contributed to a better and more rapid diagnosis, although the management remains confined to the treatment of clinical manifestations, as there are limited studies on rare diseases. Recently, the correlation between microbiota composition and the onset of autoimmune disorders has been investigated, thus opening up new perspectives on the cure of monogenic autoimmune diseases. In this review, we will summarize the main genetic features of both organ-specific and systemic monogenic autoimmune diseases, reporting on the available literature data on microbiota alterations in these patients.

## 1. Introduction

A breach of immunological tolerance is defined as the loss of the ability of our immune system to distinguish self from non-self, and it is the basis for autoimmune disease. Autoimmune diseases are believed to be mainly triggered by infections through several mechanism, such as molecular mimicry, the release of sequestered autoantigens, or the induction of inflammation (adjuvant effect); however, multiple factors, both genetic and environmental (nutrition, microbiota, and xenobiotics), can favor this breakdown, making difficult to determine the exact pathogenesis of the disease [1].

Epidemiologic studies have demonstrated that genetic factors are crucial determinants of susceptibility to autoimmune disease. Commonly, autoimmune disorders involve multiple genes, each of them exerting a minimal predisposing effect; however, rare monogenic autoimmune diseases have been identified, allowing for a better understanding of the mechanisms behind central and peripheral immune tolerance and suggesting new potential therapeutic strategies. Most of the genes causing these monogenic autoimmune diseases can alter the key mechanisms of tolerance, such as the negative selection of autoreactive T cells in the thymus or T cell apoptosis involved in switching off the immune response, or defects of T regulatory cell development or functions [1,2,3].

A contribution can also be provided by certain drugs, such as procainamide and hydralazine, by reducing DNA methylation in T cells and increasing the expression of the genes that are involved in lymphocyte activation, such as lymphocyte function-associated antigen (LFA-1), thus favoring autoimmunity; furthermore, some treatments can also alter the development of central thymic tolerance, causing the escape of autoreactive lymphocytes in the periphery [1].

Autoimmune disorders can be either organ-specific or systemic, and they may involve both adaptive and innate immune response mechanisms; although, in some cases, these differences are not so well defined.

The altered immune activation leads to tissue destruction through several mechanisms, such as the production of autoantibodies directed against surface molecules, nucleic acid, and other cellular components, which leads to cytotoxicity through antibody-dependent cell-mediated cytotoxicity (ADCC) and complements activation mechanisms, which recruit inflammatory cells, thus causing tissue damage [1]. Alternatively, autoreactive cytotoxic and helper T cells may recognize the autoantigen-derived peptides that are expressed in the target tissue and consequently induce tissue damage through the release of cytotoxic granules or the production of inflammatory cytokines (organ-specific autoimmunity). However, the role of the different types of autoreactive cells in systemic autoimmune disease is still unclear [1].

This review will focus on some of these rare monogenic diseases, such as autoimmune polyglandular syndrome (APS1), immunedysregulation polyendocrinopathy enteropathy X-linked syndrome (IPEX), autoimmune lymphoproliferative syndrome (ALPS), and monogenic systemic lupus erythematosus (SLE). In addition, we will discuss the involvement of microbiota in the pathophysiology of these monogenic diseases and the role of vitamin D in the interplay between microbiota and the onset of autoimmunity.

## 2. Microbiota and the Immune System

### 2.1. The Close Relationship between Microbiota and Immunity

Thousands of different species of microorganisms colonize the gut, the skin, and other mucosal environments in the human body. It is now known that the human microbiome plays an important role in multiple physiological functions, including metabolism, circadian rhythmicity, and shaping the immune system, contributing to health and disease [4]. In particular, microbiota promote both humoral immunity (B cell development and proinflammatory T cell responses) and immune regulation (regulatory B and T cells) [5]. A close relationship establishes between the microbiota and the immune system that continuously evolves in order to preserve the symbiosis and allows the induction of appropriate responses to pathogens and a tolerance to safe agents [5]. The first interaction between the immune system and the microbiota starts in early life, during the passage through the birth canal. Live microbes, metabolites, cells, cytokines, and immunoglobulin (Ig) A contained in maternal milk will then shape the neonatal microbiota, promoting the expansion of specific components, such as *Bifidobacterium*, or restricting immune activation against commensal antigens [6]. The establishment of microbiota is favored by an immature regulatory environment of the developing immune system. On the other hand, commensals influence early-life immunity through different mechanisms. For example, the suppression of invariant natural killer T (iNKT) cells that are involved in the inflammatory response occurs because of the interaction with inhibitory sphingolipids derived from commensals [7].

The homeostatic host/commensal relationship is fundamental and is guaranteed by the large subsets of immune cells that are resident in the sites that are colonized by commensals, such as the skin and the gastrointestinal (GI) tract. Here, the contact between microorganisms and the epithelial cell surface is minimized by mucus and the cell surface glycocalyx in order to avoid tissue inflammation and microbe translocation [8,9]. The induction of commensal-specific T regulatory cells (Tregs) preferentially occurs in GI in order to maintain the tolerance to commensal antigens and other oral-intake antigens [10]. Hence, commensals induce lamina propria resident, not inflammatory macrophages, and the local expansion of Tregs, which further promote IgA class-switching in an antigen-specific manner [11].

Interestingly, commensal-derived molecules, such as the polysaccharide A (PSA) produced by *Bacteroides fragilis*, could also induce regulatory responses. Specifically, PSA induces IL-10-producing Tregs, through the engagement of toll-like receptor (TLR) 2 on T cells, which further inhibits T helper (Th) 17 activity [12]. Besides the TLRs, additional pattern-recognition receptors (PRRs) alter the gut microbiota, such as NOD-like receptor (NLR) 2, which avoids small intestine inflammation by inhibiting the growth of the commensal *Bacteroides vulgatus* [13]. Some of the NLRs are part of multiprotein complexes, termed inflammasomes, such as for NLRP6, whose impact on the microbiota depends on the configuration of the microbiome itself, and its inflammation signaling, which affects IL-18 production by epithelial cells and, in turn, the antimicrobial peptide (AMP) expression profiles, which are influenced by microbiota-derived metabolites such as taurine, histamine, and spermine [14]. Accordingly, mouse models have shown that commensal metabolites coming from the fermentation of indigestible fibers, such as short-chain fatty acids (SCFA), can directly affect the activity of immune cells, contributing to arthritis, allergy [15], and the extrathymic generation of Tregs [16]. In addition, SCFAs promote the memory potential of antigen-activated CD8+ cells [17].

Commensal microbes control the production of cytokines, such as IL-1β and IL-18, which enhance the innate immune response against infections [18].

Among the complexity of microbiota, some species have recently emerged with a dominant role in the immune response, such as segmented filamentous bacteria (SFB) and Gram-positive anaerobic bacteria, which promote the accumulation of Th1 cells and atypical non-inflammatory Th17 cells in the small intestine, thus inducing IgA secretion [19,20]. Of note, distinct Th17 populations are supported by distinct microbes, since *Citrobacter* induce typical Th17 cells producing high amounts of pro-inflammatory cytokines [21].

The gut microbiota also interact with extra-intestinal sites, thus modulating the immune response in other organs [22,23]. Indeed, the gut microbiota and the associated metabolites can translocate through the circulatory system to different organs (such as the liver, the brain, the lungs, and even secondary lymphoid tissues), where they are recognized by local sensors and regulate the immune responses [22,23].

Overall, these data suggest the existence of a complex network of bidirectional interactions between microbiota and both innate and adaptive immunity cells, which is fundamental to prevent autoimmune reactions and a high inflammatory status with consequent tissue injury. Since the immune system and the microbiome are tightly interconnected in a complex network, the growing interest in the possible role of the microbiome in autoimmune diseases is not surprising.

### 2.2. The Role of Vitamin D in the Interplay between Microbiota and the Immune System

In the context of microbiota and the immune system interconnection, vitamin D has recently emerged as one of the regulators of this complex network [24].

Interestingly, vitamin D shortage has been often reported in several autoimmune diseases, such as type I diabetes mellitus, systemic lupus erythematosus, rheumatoid arthritis, and multiple sclerosis, and it is suspected to play a role in the pathogenesis of these diseases [25,26,27]. A lack of vitamin D may in fact impair the integrity of the gut epithelial barrier and, consequently, the microbiota balance [28]. Vitamin D/vitamin D receptor (VDR) signaling normally enhances the expression of several components of tight junctions and adherent junctions, such as occludin and claudins, in the intestinal epithelium [29]. The loss of such epithelial junctional proteins due to vitamin D deficiency could contribute to “leaky gut” syndrome, which favors the absorption of microbic components, alters the microbial composition, and promotes immune cell activation. A meta-analysis on patients with inflammatory bowel disease has interestingly indicated that the correction of hypovitaminosis D contributes to better maintenance and reduces the risk of disease relapse [28,30].

VDR is a nuclear receptor that can be found in about 30 different tissues, and it is able to regulate the expression of more than 1000 genes in the genome, after binding to the active form of vitamin D, 11α,25(OH)2D3 [31]. Studies have identified four single nucleotide variants (SNVs) in the VDR gene, named *FokI* in exon 2, *BsmI* and *ApaI* in intron 8, and *TaqI* in exon 9, which affect the structure of the receptor and, consequently, the response to vitamin D, and seem to be involved in autoimmune disease [32,33].

Within the immune system, 1α,25(OH)2D3 modulates the immune response via the VDR expressed in naïve and CD4+ and CD8+ T cells, B cells, neutrophils, and antigen-presenting cells, such as monocytes, macrophages, and dendritic cells [34]. Vitamin D promotes Tregs, inhibits the differentiation of proinflammatory Th1 and Th17 cells, impairs the development and the function of B cells, modulates the regulatory B cells, reduces monocyte activation, downregulates IL-17 and IL-23 production, and promotes IL-10 production [34]. Since Th1, Th2, and Th17 cells can support mucosal inflammation and tissue injury, while Tregs are important players of immune tolerance, as they exert anti-inflammatory functions and stimulate tissue repair [35], vitamin D substantially influences the immune response by balancing inflammation versus anti-inflammation, and vitamin D deficiency disrupts this balance, favoring inflammation [36].

Vitamin D and microbiota display many similarities in the modulation of the immune system, downregulating the proinflammatory pathways and the inflammatory markers, and upregulating anti-inflammatory cytokine synthesis [35]. These similarities could be due to the synergistic effect between vitamin D and microbiota metabolites such as SCFAs. In fact, both can exert anti-inflammatory activity by promoting Treg function, increasing the levels of the anti-inflammatory cytokine IL-10, affecting the maturation of DC, and inhibiting the expression of pro inflammatory cytokines [35] (Figure 1).

In addition, human studies have reported significant associations between vitamin D and the microbiome genus composition [37]. A recent genome-wide association study has identified two VDR polymorphisms significantly influencing the abundance of the genus *Parabacterioides* (phylum: Bacteroidetes), which has been confirmed in VDR−/− mice showing an increased abundance of *Parabacteroides* compared to wild-type mice [33].

In conclusion, vitamin D appears to be a critical player in the dynamic relationship between the immune system and the microbiome, with predominantly immunosuppressive properties; therefore, vitamin D supplementation has been suggested to be of benefit in the treatment of autoimmune diseases. However, little is known about how vitamin D supplementation impacts the microbiome in autoimmune patients since few studies have addressed this question [37].

In the following section, we analyze both organ-specific and systemic monogenic autoimmune disorders, summarizing their genetic features, the involvement of microbiota, and the available data on the role of vitamin D in their pathogenesis in order to provide a comprehensive review and potential suggestions for further studies.

## 3. Pathophysiology of Organ-Specific Monogenic Autoimmune Disorders and Microbiota Implication

### 3.1. Autoimmune Polyendocrine Syndrome Type 1 (APS1)

Autoimmune polyendocrine syndrome type 1, which is also known as autoimmune polyendocrinopathy candidiasis-ectodermal dystrophy (APECED), is a rare recessive disorder that mainly manifests with mucositis (from the mouth to the gastrointestinal tract), hypoparathyroidism, adrenal insufficiency, and displays a great variability in the timing of the onset [38]. However, other various symptoms have been described, such as alopecia, vitiligo, premature ovarian failure, and type I diabetes [39].

The autoimmune regulator gene (AIRE) was identified in 1997 as the causal gene on chromosomal region 21q22.3 by positional cloning [40,41]. AIRE encodes a 545 amino acid protein that contains several domains that are characteristic of transcription factors, including a homogeneously staining region (HSR) domain (involved in dimerization), followed by a SAND domain (DNA binding), and two plant homeodomain (PHD) domains (probably E3 ubiquitin ligase activity) [42].

AIRE is mainly expressed in the thymus (medullary thymic epithelial cells (mTEC) and dendritic cells (DCs)), and, to a lesser extent, in secondary lymphoid organs. It regulates the ectopic expression of tissue-specific proteins, thus contributing to the deletion of autoreactive T cells by both central and peripheral tolerance [43,44].

More than 60 different mutations have been described, and the most frequent is the 1094-1106del13 [45]. Generally, heterozygous patients for other APS1 mutations do not develop the typical clinical features of APS1, but may manifest different uncommon phenotypes. A G228W pathogenic variant with dominant inheritance was discovered in an Italian family, displaying a nonclassical autoimmune thyroiditis [46,47]. Overall, a low correlation between the genotype and the phenotype has been reported and, even in siblings, the same mutation may cause different clinical manifestations [48].

Defects in the number and the function of Tregs characterize APS1 patients, as well as abnormal DCs [49,50]. In addition, the infiltration of lymphocytes in affected organs (the stomach, lungs, and eyes), as well as the production of autoantibodies (autoAbs) against antigens (i.e., insulin and cytochrome p450 1A2) of the affected tissues, have been described [51]. The early production of high titer of autoAbs against interferon (IFN)-ω and -α2 are also used in diagnosis [52]. More recently, the NACHT leucine-rich repeat protein NALP5 (also known as NLRP5) has been identified as a putative parathyroid autoantigen to be evaluated in patients with hypoparathyroidism [53].

Furthermore, autoAbs to interleukin (IL)-17A, IL-17F, and IL-22 have been detected in the sera of all APS1 patients with candidiasis, thus suggesting an impaired role of the Th17 cells in fungal immunity [54]. In addition, a recent study showed AIRE interaction with Dectin-1, which is involved in the inflammasome-activating pathway during the anti-fungal response, which suggests a possible impairment of inflammasome activity in APS1 patients [55].

Among the clinical manifestations, gastrointestinal symptoms frequently occur in APS1 patients [56]. Studies have demonstrated that they correlate with the lack of tolerance to self-antigens such as tryptophan hydroxylase and histidine decarboxylase, which are produced by the enteroendocrine cells (EECs) of the gut epithelium [57]. The consequent loss of EECs causes low serum levels of serotonin and typical constipation symptoms [57].

Interestingly, APS1 patients show serum reactivity against the secretory granules of Paneth cells containing enteric α-defensins and the loss of Paneth cells themselves, together with the infiltration of T cells in the gut [58]. Enteric defensins have antimicrobial activity and influence the ecology of the commensal microbiota [59]. Specifically, a study by Petersen A. et al. [60] analyzed the fecal microbiomes of APS1 patients and found a correlation between several taxa, mainly belonging to the *Clostridiales*, and gastrointestinal symptoms. Notably, the depletion of a commonly used probiotic *L acidophilus* correlated with an increased Bacteroidetes–Firmicutes ratio in APS1 patients [60]. Moreover, Hetemaki I et al. [61] reported high levels of Abs against *Saccharomyces cerevisiae* (S. cerevisiae) and several species of commensal gut bacteria, which did not correlate with gastrointestinal autoAbs and neutralizing anti–IL-17 or –IL-22 autoAbs, or gastrointestinal symptoms. Conversely, anti–S. cerevisiae Ab levels were inversely correlated with the Foxp3 expression levels in Tregs, which are crucial for the maintenance of mucosal tolerance and are numerically decreased in the gut biopsies of APS1 patients [61]. The Treg defect may allow the resident T and B cells to mount a response against commensals supported by the defects in the Th17 cytokines. Of note, a shift toward an immunoglobulin (Ig) G response has been described in these patients, in contrast with the local IgA response, which characterizes the normal gut with normal flora [61]. Anti–S. cerevisiae Abs have been also associated with Crohn’s disease [62], thus suggesting that AIRE plays a role in regulating gut homeostasis.

Studies on Foxp3-deficient *scurfy (SF) mice* have shown that autoimmunity is accompanied by decreased gut bacterial diversity, which is restored by Foxp3^+^ Tregs administration [63]. Moreover, the treatment with *Lactobacillus reuterii* (DSM 17938, LR 17938), a human-derived probiotic, which has been used to treat infantile colic and acute infectious diarrhea, repristinated the flora heterogeneity and decreased the disease severity [64]. Nevertheless, whether dysbiosis is a response to or a driver of the gastrointestinal symptoms that are characteristic of APS1 remains to be determined.

Currently, APS1 treatment is based on targeting organ-specific manifestations. Antifungals are used in the case of candidiasis as a replacement therapy for endocrinopathies, with cortisol and fludrocortisone being given for adrenal insufficiency, and calcium and vitamin D as a replacement for hypoparathyroidism [39]. The deficit of vitamin D has been described in several patients with APS1 who also showed severe hypocalcemia [65]. The authors suggest that the loss of EECs in the gut epithelium contributed to vitamin D malabsorption, making useless the oral administration of calcitriol, while suggesting a better efficacy of parental vitamin D analogs in controlling hypoparathyroidism in these patients [65,66]. In light of the role of vitamin D in autoimmunity and its effects on microbiota, it is conceivable that its deficit in APS1 patients might be also involved in the pathogenesis of the disease or correlated with the severity of the clinical manifestations. However, studies on the involvement of vitamin D in APS1 are currently missing, thus opening interesting new research fields.

### 3.2. Immunodysregulation Polyendocrinopathy Enteropathy X-Linked (IPEX) Syndrome

IPEX is a rare X-linked recessive disorder, first described in 1982 [67], resulting in early onset type 1 diabetes mellitus (T1DM), severe enteropathy, eczema, anemia, thrombocytopenia, hypothyroidism, and a hyperreaction to viral infections [68]. If it is not diagnosed early, IPEX has a very severe onset and is usually lethal in infancy or childhood, independently from the type of causative mutation. Patients frequently show high levels of IgA and IgE and eosinophilia [68]. Furthermore, autoAbs targeting pancreatic islets, thyroid, erythrocytes, platelets, and the small intestine are frequently detected [68].

In 2000 [69], IPEX was mapped to the Xp11.23-Xq.13.3 chromosomal region where *FOXP3* (Forkhead box protein 3) was identified as the responsible gene.

*FOXP3* is a highly conserved gene that is composed of 12 exons, encoding the 431 amino acid protein Scurfin, which is the master transcription factor for Treg functions, is involved in the expression of the main markers of these cells, such as CTLA-4 and CD25, and is responsible for the suppression of proinflammatory cytokine production [70]. Over 70 pathogenic variants have been reported so far, mainly at the C-terminal forkhead (FKH) DNA-binding domain. Mutations of the polyadenylation site are causative of unstable FOXP3 mRNA, resulting in severe and early onset disease [71]. However, the severity of the disease is not strictly dependent on the absence of the protein, and most of the mutations (missense) cause an expression of normal or reduced levels of mutant protein. Normal percentages of CD4^+^CD25^+^FOXP3^+^ T cells can be detected in the peripheral blood of IPEX 1 patients; however, these mutant FOXP3 Tregs are functionally impaired and are the main pathogenetic cause of the multi-organ autoimmunity in IPEX patients [72]. In vitro studies have described a high heterogeneity in the severity of functional impairments among Tregs from different IPEX patients [72]; moreover, a lack of correlation between the genotype and the phenotype has been reported, which reflects the complexity of FOXP3 interactions and the role of non-genetic factors in the clinical manifestations of the disease, such as environmental and epigenetic factors [73]. In addition to the loss of the suppressive functions, an inflammatory shift toward the Th17 phenotype has been reported in patients with mutant FOXP3 Tregs, which might be involved in organ damage [74]. Furthermore, a decreased production of Th1 cytokines and increased levels of Th2 cytokines have been shown by several authors [72,75,76], which suggests that *FOXP3* mutations may also affect T effector cell activities.

At least 50% of IPEX patients do not carry mutations in *FOXP3*. In children who are diagnosed with an IPEX-like syndrome, the genetic defect is either unknown or alterations in other genes that are related to Treg functions have been detected, such as in *IL2RA, STAT 5b, ITCH, STAT1, STAT3, CTLA4, LRBA, TTC7A*, and *TTC37* [77].

The therapeutic strategies depend on the clinical manifestations and include replacement and supportive therapy targeting the damaged organs, immunosuppressive agents, and hematopoietic stem cell transplantation (HSCT) [73]. Early HSCT can lead to the best outcome when autoimmunity has not yet damaged the organs, hence making an early diagnosis fundamental. A future cell/gene therapy approach may represent an option to replace mutant cells with wild-type Tregs, which seem to be sufficient to control the disease. A recent study has suggested a CRISPR-based gene correction that can specifically target FOXP3 in autologous hematopoietic stem and progenitor cells (HSPCs). This approach allows us to edit HPSCs while preserving their differentiation potential and represents a promising strategy for IPEX patients with different types of mutations [78].

Fecal microbiota transplantation (FMT) has been suggested as an alternative strategy to cure IPEX patients with enteropathy symptoms [79]. Wu et al. specifically described a significantly decreased diversity in the gut microbiota composition in a child with IPEX, which was repristinated after receiving FMT along with remission of diarrhea [79]. A correlation between gut dysbiosis and gut inflammation has been previously reported in the IPEX model FoxP3^+^ Treg cell-deficient *SF mouse* [80,81]. Indeed, these studies showed significant differences in the microbiota composition in 22-day-old SF mice as compared with wild-type (WT) mice by analyzing stool 16S rRNAs. Specifically, the SF mice showed a decreased relative abundance of *Lactobacillus* at day 8 and increased *Bacteroides* at day 22, thus suggesting a role of Treg deficiency in shaping the gut microbiota composition [64]. Moreover, the treatment with 10^7^ CFU/day of *Lactobacillus reuterii*, which was orally fed by gavage, reset SF dysbiosis, increasing the relative abundance of the phylum *Firmicutes* and the genera *Lactobacillus* and *Oscillospira*, and decreasing the relative abundance of the phylum *Tenericutes* and the genus *Bacteroides* [64].

In addition to remodeling the gut microbiota, the treatment with *Lactobacillus reuterii* significantly increased the survival of the SF mice, which usually die after 21 days. It reduced the T lymphocyte infiltration in the lungs and the liver and decreased the IFN-γ and IL-4 plasma levels. This indicates that *Lactobacillus reuterii* also suppresses the autoimmunity that is caused by Treg deficiency [64]. An analysis of the plasma metabolites also demonstrated significant effects on alterations that were correlated to Treg deficiency; for example, the immunomodulatory purine metabolite inosine, but not adenosine, hypoxanthine, or xanthine, was reduced 5-fold in the SF mice and the treatment with *Lactobacillus reuterii* restored these levels [64]. In addition, decreased multiorgan inflammation and prolonged survival was described after inosine feeding, which inhibited the Th1/Th2 differentiation in an adenosine A_2A_-receptor-dependent manner [64].

IPEX dysbiosis was also restored by the administration of Foxp3 + Tregs in T-cell-deficient mice [63].

Overall, these data suggest multiple pathways that could be targeted in order to treat the disease, and respective therapeutic strategies that could be investigated.

## 4. Pathophysiology of Systemic Monogenic Autoimmune Disorders and Microbiota Implication

### 4.1. Autoimmune Lymphoproliferative Syndrome (ALPS)

ALPS is a disorder of T cell-apoptosis dysregulation that leads to chronic polyclonal, but not neoplastic lymphoproliferation, autoimmune manifestations and an increased development of lymphomas [82,83,84].

ALPS patients show multiple manifestations, such as lymphadenopathy, splenomegaly, hypergammaglobulinemia, and autoimmune features that are often represented by hemolytic anemia and thrombocytopenia, but also hepatitis, uveitis, and vasculitis [84]. Systemic autoimmunity frequently requires medical intervention, as the destruction of several blood cell lines often occurs, leading to chronic cytopenia, which can range from mild to severe, and is treated with immunosuppression [82]. Autoimmunity usually manifests in early childhood, months or years after lymphoproliferation.

ALPS was first described in the 1990s in a cohort of patients with chronic lymphoproliferation and an increased number of double negative T cell populations (DNTs; cell phenotype CD4^−^/CD8^−^, CD3^+^, TCRαβ^+^), which reach <1% in the peripheral blood of healthy controls (vs. >2.5% in ALPS patients) [85]. Increased DNTs in the peripheral blood and the lymphoid tissues is a hallmark of ALPS, together with defective in vitro Fas-mediated apoptosis. However, mild increases in DNTs have been also described in other autoimmune diseases, including SLE and immune thrombocytopenic purpura [86,87]. The origin of DNTs is still not clear, though a significant overlap of TCR Vβ-Jβ transcripts between DNTs and CD8 T cells has emerged from CDR3 sequencing in ALPS patients, thus suggesting at least a partial CD8 origin [88,89].

Elevated serum levels of IL-10, vitamin B12, and Fas ligand have been also reported in ALPS patients [90,91].

As for IPEX and APS1, mouse models with MRL*lpr/lpr* and MRL*gld/gld* mice with similar clinical features have contributed to a deep understanding of the diseases, also highlighting the importance of apoptosis in the maintenance of immunologic homeostasis and tolerance. These mice carry loss-of-function mutations of either the FAS (*lpr* mutation) or the FASLG (*gld* mutation) genes, causing defective FasL/Fas-induced apoptosis [92,93]. Pathogenic variants in these genes are detected in approximately 70% of ALPS patients and are usually inherited with autosomal dominant mechanisms [94]. These mutations alter the formation of the Fas/FasL functional trimolecular complex, which is required to induce apoptosis in effector lymphocytes and controls their lifespan [95]. As a consequence, lymphocytes accumulate in the secondary lymphoid tissues, which favors development of autoimmunity and, possibly, lymphomas.

The most frequent mutations that hit FAS and are often characterized by incomplete penetrance, and the inheritance pattern can be dominant with incomplete penetrance, recessive, or can be manifested when it is associated to variants in other genes, thus suggesting a possible digenic/oligogenic inheritance [96,97]. ALPS that is caused by germline mutations of FAS is classified as ALPS-FAS, but some cases are caused by somatic mutations of this gene (ALPSsFAS). Rare cases are due to mutations of FASL (ALPS-FASL) or CASP10 (ALPS-CASP10) that are involved in Fas signaling, whereas the mutated gene is undetermined in about 30% of cases (ALPS-U) [98]. Some studies report that mutations at the heterozygous state in the FAS intracellular domain affect the canonical death domain that is required for initiating the apoptotic cascade and are associated with a higher penetrance, compared with heterozygous mutations of the extracellular domain [99]. However, there is no consensus about the penetrance of the disease and its correlation with the type of mutation, thus suggesting a recommendation of genetic counselling for all of the unaffected family members who inherited the pathogenic variant.

In addition, it may be difficult to diagnose ALPS, since a biological and clinical overlap exists with other diseases, which may show lymphoproliferation, autoimmunity, and apoptosis defects, such as caspase-8 deficiency (CEDS), RAS-associated autoimmune leukoproliferative disease (RALD), Dianzani autoimmune lymphoproliferative disease (DALD), and even X-linked lymphoproliferative syndrome (XLP) [84,86,100,101,102,103]. Moreover, a defective Fas function in the absence of FAS mutations can be detected in subsets of patients with common autoimmune diseases such as multiple sclerosis, type I diabetes mellitus, and SLE [104,105,106].

Furthermore, a tissue biopsy (bone marrow and/or lymph node) is recommended at the initial presentation in order to distinguish ALPS from malignancy [107].

More recently, a role of antigen-presenting cells (APCs) has been suggested in the onset of ALPS, since defective apoptosis of APCs has been reported in ALPS patients [108]. Moreover, studies on murine models describe autoimmunity signs after conditional inactivation of Fas in APCs (DCs or B lymphocytes), with the production of high levels of antinuclear Abs (ANA), hyper-immunoglobulinemia, and splenomegaly, without DNT expansion [109]. These findings suggest that chronic antigen presentation, from either activated DCs or B cells with defective apoptosis, can also disrupt immune tolerance.

To our knowledge, no data are currently available on the involvement of microbiota in ALPS patients, leaving space for future research.

ALPS patients may not require any treatment, but rather clinical surveillance. In the presence of clinical manifestations, immunosuppressive agents are used to treat autoimmune manifestations and chemotherapy is used for malignancy [107].

### 4.2. Monogenic Forms of SLE

SLE is a complex multisystem autoimmune disease with heterogeneous clinical manifestations. The pathogenesis is correlated to different genetic backgrounds, immunologic alterations, and environmental factors, which makes it difficult to fully understand the SLE mechanisms and, consequently, to develop therapeutic strategies [110,111].

However, the identification of a rare monogenic form of SLE, characterized by familial segregation, as well as early-onset juvenile SLE, allowed for a better understanding of the involved pathways, i.e., complement, apoptosis, nucleic acid degradation, nucleic acid sensing, self-tolerance, and type I IFN production [112].

#### 4.2.1. Genetic Alterations

##### The Complement System

The complement system plays a key role in the innate and adaptive immune response. Some components act as *opsonins*, which allow the clearance of immune complexes by red blood cells and phagocytes, thus avoiding tissue vascular deposition. Moreover, early components of the classical pathway (C1q, C1r, C1s, C2, and C4) decrease the release of autoantigens by promoting the clearance of apoptotic cells by phagocytes (efferocytosis) [113]. Its role in T and B cell activation has been described as well, which may alter the balance of lymphoid cell activation [114,115]. C1q also reduces type I IFN production by plasmocytoid DCs [116] and modulates CD8+ T cell mitochondrial metabolism in order to avoid self-antigen autoreactivity and promote viral control [117].

Complement C1q, C1s, and C1r deficiencies are associated with early-onset SLE, although they are very rare (<1%). Most of these patients show skin involvement and a severe SLE phenotype, together with recurrent life-threatening infections, such as meningitis; moreover, high levels of ANA and rarely anti-double-stranded DNA (dsDNA) have been described [118]. SLE patients with homozygous C2 deficiencies are characterized by severe arthritis and cutaneous vasculitis, while nephritis is frequent in those patients with C4 defects [119].

Dysregulations of apoptosis, as well as NETosis, i.e., the release of neutrophil extracellular traps (NETs), are also thought to be involved in monogenic SLE, increasing the autoantigen load and disrupting tolerance [120]. As mentioned previously, ALPS, FAS, and FASL mutations are implicated in the development of autoimmune responses, which may mimic SLE in some patients. MRL*lpr/lpr* mice have been often used as models of SLE [121,122].

##### PRKCD

A study by Belot et al. [123] identified PRKCD mutations in three siblings with juvenile-onset SLE. *PRKCD* encodes protein kinase C (PKCδ), which is involved in the deletion of autoreactive B cells [124]. The mutation causes a reduction in PKCδ expression and activity, thus leading to B cell expansion and defective apoptosis [123]. More recently, other studies have supported *PRKCD* as a candidate gene that is associated with SLE risk [125,126]. Monogenic pathogenic variants in *RAG2* (recombination activating 2 gene) that are involved in V(D)J recombination and BCR to TCR variability generation also cause SLE manifestations, including lupus nephritis and erosive arthritis [127].

T cell expansion and defective apoptosis also occur in some patients with Noonan syndrome, a neurodevelopment disorder that is caused by pathogenic variants in one of the several RAS/MAPK pathway genes, including *PTPN11, SOS1, RIT1*, and *KRAS*, encoding for a small GTPase that is activated by TCR signaling [128,129]. This disease manifests with a short stature, dysmorphic features, congenital heart disease, bleeding diathesis, and an increased risk of malignancy and lupus-like symptoms [130].

##### TREX1

The degradation of nuclear materials is another mechanism that prevents autoantigen load. Several intracellular enzymes, such as three prime repair exonuclease 1 (TREX1) and RNase H2 complex, are involved in this function. *TREX1* encodes an IFN-inducible exonuclease that is responsible for genomic DNA degradation in response to DNA damage. Therefore, it is involved in the immune response to single-stranded (ss)-DNA and dsDNA and in the maintenance of immune tolerance to cytosolic self-DNA [131]. Dysfunctional *TREX1* causes an accumulation of self-DNA, which represents a damage-associated molecular pattern (DAMP), activating systemic inflammation and triggering autoimmunity [132]. Similarly, RNase H2 degrades RNA, DNA heteroduplexes and contributes to the ribonucleotide excision repair pathway [133].

Alterations of genes encoding for these enzymes have been reported in Aicardi–Goutières syndrome (AGS), an autosomal recessive disease affecting the skin, the brain, and the immune system, which phenotypically overlap with SLE [134]. Biallelic mutations of *TREX1* cause a complete loss of protein function in AGS patients, who show high levels of IgG autoAbs directed against the nuclear antigens, the basement membrane components, the gliadin and brain endothelial cells, and the astrocytes [135]. Partial loss-of-function biallelic mutations in RNase H2 also occur in AGS, leading to an intracellular accumulation of aberrant nucleic acid species and the consequent activation of nucleic acid sensor and IFN cascade [136].

The loss-of-function mutations of *TREX1* (also called DNAse type III) and genes of the RNase H2 complex have been found in SLE patients as well [134,137,138].

AGS patients also show high levels of type I IFN in the serum and the cerebrospinal fluid [139] because of the mutated TREX1 enzyme, which is not able to prevent intracellular DNA detection by innate immune sensors, thus inducing type I IFN activation and accumulation. The excessive production of type I IFNs is the hallmark of SLE too [140]. This signature induces immature DCs towards an effector phenotype (APCs), suppresses Tregs, increases the differentiation of Th17 cells, and stimulates B cell proliferation [141].

##### DNASE1L3

A study by Al-Mayouf SM et al. [142] identified another gene, *DNASE1L3*, which was mutated in familial cases of juvenile SLE, with anti-dsDNA reactivity. DNASE1L3 is an extracellular DNA-degrading enzyme, a homologue of DNASE1, with a positively charged C-terminal peptide, which confers the ability to remove DNA chromatin that is associated to the extracellular microparticles [142]. It has been recently shown, indeed, that, in these patients, DNA accumulates in the circulating microparticles containing high levels of long polynucleosomal cell-free DNA (cfDNA) fragments, with an increased affinity to auto-Abs and capacity to induce type I IFN production after recognition by DNA/RNA sensors such as Toll-like receptors (TLR7 and TLR9) [143]. The amount of auto-Abs against DNASE1L3-sensitive antigens on microparticles correlates with disease severity and is frequent in SLE nephritis patients [143]. Among these antigens, the authors describe the high-mobility group B1 (HMGB1) protein, belonging to structural transcription factors, that can be incorporated into chromatin or released from activated or dying cells [143]. Pisetsky D et al. [144] previously described the exposure of HMGB1 on apoptotic microparticles, while Hartl J and colleagues [143] demonstrated that DNASE1L3 controls this phenomenon through the digestion of HMGB1–DNA complexes, overall suggesting a new potential therapeutic target in the monogenic forms of SLE [143]. Mutations in *DNASE1* have also been described in patients with SLE manifestations and high titer of ANA [145].

##### ACP5

Another disease that is usually associated with autoimmune SLE is spondyloenchondrodysplasia (SPENCD), a rare autosomal recessive skeletal dysplasia caused by mutations in the *ACP5* (acid phosphatase 5) gene [146,147]. *ACP5* encodes tartrate-resistant acid phosphatase (TRAP), which uses osteopontin (OPN) as a primary substrate [148]. OPN is also required for type I IFN production in response to TLR9 stimulation by plasmacytoid DCs [149]. *ACP5* mutations alter the TRAP activity, causing OPN hyperphosphorylation, which in turn induces excessive type I IFN production and SLE-like pathologies [147].

Overall, the monogenic forms of SLE provide opportunities for a better understanding of the mechanisms behind the disease and allow for the design of precise therapeutic strategies. SLE patients with an IFN signature could benefit from a human anti-IFN Ab, which competes with IFN-α binding to the receptor IFNAR2, thus preventing the activation of the IFN-α mediated pathways [150]. Interestingly, a systematic literature review by Rodríguez-Carrio et al. [151] has revealed that measuring type I IFN pathway activation in these patients could potentially help in monitoring the disease activity, the prognosis, and the response to treatment, although assay harmonization and clinical validation are still needed [151].

Since nucleic acid sensors, such as the retinoic acid inducible gene (RIG)-1 and the melanoma differentiation-associated gene 5 (MDA5), activate type I IFN via TBK-1 signaling, the use of TBK inhibitors might represent an alternative strategy in SLE patients, with fewer side effects than the conventional immunosuppressive drugs [152]. The reverse transcriptase inhibitors that are currently used to treat human immunodeficiency virus (HIV) could be also used in monogenic forms of SLE with mutated TREX1 or RNase H2 complex, which alter the metabolism of the nucleic acids that are generated during reverse transcription [153].

The use of NGS (next-generation sequencing) in early-onset SLE will provide further insights in understanding the molecular mechanisms of the disease, but also of tolerance maintenance, leading to a personalized medicine with less adverse events and increased efficacy [154].

#### 4.2.2. Microbiota Alterations

Due to the extensive involvement of multiple organs, a higher level of fatigue is observed in patients with SLE who also show a deficit in vitamin D, which worsens the condition [155,156]. Several studies have explored the association between calcidiol (the main vitamin D metabolite) serum levels and the disease activity and susceptibility, although the results are still inconclusive, because of the wide calcidiol range that has been reported in the different analyzed populations. Indeed, its levels are influenced by environmental and genetic factors, making it difficult to compare different studies [32,155,157]. Moreover, high levels of the active form, calcitriol, have been detected in SLE patients and could be indicative of active inflammation and Th2 polarization [156].

Several recent studies, on both mouse models and humans, suggest a correlation between alterations of the gut microbiota composition and SLE disease manifestations [158]. The main feature is that the ratio of Firmicutes/Bacteroides (F/B) is significantly reduced, and the decreased abundance of some families of Firmicutes may be related to remissive SLE [159]. *Synergistetes* negatively correlate with anti-dsDNA Abs [160]; meanwhile, anti-*Ruminococcusgnavu*s Abs positively correlate with lupus nephritis and anti-dsDNA Abs [161]. A positive correlation was found between *Streptococcus*, *Campylobacter*, *Streptococcus anginosus*, and *Veillonella dispar* and lupus activity, while Bifidobacterium was negatively associated [162]. Low levels of *Lactobacillus* and increased *Lachnospiraceae*, with inflammatory properties, have been described. Gender differences in the gut microbiota of SLE mice have also emerged [159].

Interestingly, microbiota isolated from SLE patient stool samples induce lymphocyte activation and Th17 differentiation from naïve CD4+ lymphocytes [160].

These data suggest that the modulation of gut microbiota via diet, probiotics/prebiotics, and FMT might be a promising additional therapeutic strategy for SLE, possibly also including the monogenic forms of SLE.

### 4.3. The Role of Microbiota in Murine Models of Systemic Autoimmune Disorders

As mentioned previously, studies on murine models have helped to clarify the mechanisms behind the microbiota involvement in monogenic autoimmune diseases. Comparable results have been reported from different groups, using well-established mouse models that show similar features of both ALPS and SLE patients, as follows: MRL/lpr, which spontaneously develops autoimmune disease, with hypergammaglobulinemia, the production of numerous ANA, anti-dsDNA, circulating immune complexes, lymphadenopathy, and glomerulonephritis; and MRL+/+, with a similar background, but with a much slower onset of SLE. Wang et al. [163] used 6- and 18-week-old mice and highlighted several differences in terms of the microbial composition, the intestinal barrier integrity, and the inflammatory markers among the models. Specifically, a lower *Firmicutes/Bacteroidetes* ratio was reported in the 6-week-old MRL/lpr mice compared to the age-matched MRL+/+ mice, with decreased levels of *Peptostreptococcaceae* (Firmicutes phylum) and increased *Rikenellaceae* (Bacteroidetes phylum) [163]. No correlation was found with the advanced disease activity in the 18-week-old mice; however, a decreased number of *Lactobacillaceae* was reported in these mice compared with their 6-week-old counterparts. Interestingly, previous studies have shown that the oral administration of a mixture of five strains of Lactobacillus spp (*L. reuteri, L. oris, L. johnsonii, L. gasseri*, and *L. rhamnosus*) restored the microbiota in the MRL/lpr mice and attenuated the disease symptoms, thus improving the renal functions, splenomegaly, and lymphadenopathy [164]. This group further investigated the mechanisms behind *Lactobacillus* effect, demonstrating the restoration of the Treg-Th17 balance, with a decreased percentage of central memory T cells and increased effector memory T cells, along with reduced IL-6 and augmented IL-10 production in the gut. Of note, the strains acted synergistically in order to ameliorate the disease in a CX3CR1-dependent manner [165]. CX3CR1 is involved in the translocation of the APCs from the blood to the mesenteric lymph nodes, where they can activate T cells, thus contributing to the homeostasis between the commensal bacteria and the immune system [166]. Furthermore, the same benefits of the mixed Lactobacilli were obtained using culture supernatants, suggesting a potential use of both probiotics and postbiotics in the management of SLE as a cost-effective approach in support of the conventional treatment strategies. In this regard, Da Som Kim et al. [167] demonstrated an improved efficacy of the immunosuppressant Tacrolimus, which was administered in combination with Lactobacillus acidophilus in MRL/lpr mice. A decreased number of double negative T cells and lower serum levels of anti-dsDNA Abs and IgG2a were detected after the treatment, along with improved renal functions [167]. A higher production of indoleamine-2,3-dioxygenase (IDO), PD-1, and IL-10 was reported after the in vitro treatment with *Lactobacillus*, and it seems to be mediated by the specific intracellular adhesion molecule-3 grabbing the non-integrin homolog-related 3 receptor (SIGN3) signals, which is a homologue of the human DC-SIGN [167]. The positive effects of *Lactobacillus* treatments were confirmed in other SLE mouse models, such as the NZBWF1 model, which showed reduced lupus disease activity, blood pressure, cardiac and renal hypertrophy, and splenomegaly after *Lactobacillus fermentum* CECT5716 (LC40) intake. Endothelial dysfunctions, which characterize SLE disease, were also solved, and decreased plasma levels of LPS were detected, suggesting an improved gut barrier integrity [168].

A study by Hanchang He et al. [169] gave evidence of dysbiosis improvement after sodium butyrate treatment in MRL/lpr mice. Butyrate belongs to the SCFA, which are the products of intestinal bacterial fermentation. It increases the expression of the tight junction proteins, contributing to the integrity of the gut barrier, and it reduces oxidative stress and mucosal inflammation. In the MRL/lpr mice, the butyrate treatment enhanced the gut microbiota diversity, increasing the abundance of *Firmicutes, Clostridia, Clostridiales*, and *Lachnospiraceae*, while reducing *Bacteroidetes, Bacteroidia*, and *Bacteroidales* [169]. In addition, the promoting effect of butyrate on fat oxidation through brown adipose tissue activation [170] further supports its beneficial activity on SLE, according to the efficacy of caloric restriction in preventing disease progression that has been shown in NZB and (NZBxNZW) F1 mice, which spontaneously develop lupus, slowly.

High levels of oxidative stress molecules, such as 4-hydroxynonenal (HNE)-adducts and HNE specific immune complexes, were also detected in MRL/lpr mice, similarly to human SLE [171]. Protein oxidation or lipid peroxidation can affect the intestinal permeability in vitro and in vivo, thus worsening the already altered gut barrier, as shown by the decreased expression of the tight junction protein ZO-2, the increased fecal albumin, and the IgA levels that have been described in 18-week-old MRL/lpr mice [171]. Inflammatory markers such as phos-NFkB, IL-6, and IgG levels also characterized these mice. Intriguingly, the treatment with a ROS scavenger, N-acetylcysteine, not only downregulated the hepatic inflammasome activation markers, but also changed the microbiota composition in these mice by reducing *Rikenellaceae*, while supporting *Akkeransiaceae, Erysipelotrichaceae*, and *Muribaculaceae* colonization, finally attenuating autoimmunity [171].

Additional studies have investigated the involvement of the gut microbiota in the therapeutic response to glucocorticoids in SLE. Prednisone upregulated the concentration of *Proteobacteria, Parabacteroide*s, and *Escherichia Shigella* [172]. Moreover, experiments of FMT from prednisone-treated MRL/lpr mice revealed that the altered microbiota after the glucocorticoid treatment recapitulate the therapeutic effects of the glucocorticoid itself on SLE, without showing side effects [172].

All together, these data support the involvement of the gut microbiota within SLE and autoimmune lymphoproliferative disease progression, thus suggesting a new potential therapeutic strategy, which should be further investigated.

## 5. Conclusions

The study of monogenic autoimmune diseases has importantly contributed to our understanding of the immune tolerance mechanisms.

The complex causes of autoimmune diseases present a challenge to the development and testing of novel prognostic markers and therapies. Mouse models have provided data on the molecular mechanisms of autoimmunity and have suggested new potential therapeutic approaches. For example, the adoptive transfer of Tregs for the treatment of type I diabetes has emerged from mouse studies, and several groups are moving toward using this strategy even in other diseases, such as graft vs. host disease [173]. Likewise, the diagnosis of more common diseases, such as interstitial lung disease or autoimmune gonadal failure, has been facilitated by the discovery of autoAbs in APS1 patients [174]. Overall, these data highlight the importance and the contribution of studying monogenic autoimmune disease in the diagnosis and treatment of common autoimmune diseases.

In addition, recent advances in deep sequencing have disclosed new defective genes in isolated rare families, thus providing new insights into the development of future personalized therapies [46]. Genetic testing, together with laboratory biomarkers and autoimmune features, is crucial to accurately diagnose monogenic diseases (see Table 1 for summarized disease features); however, it remains challenging, due to the variability in disease progression and overlapping clinical manifestations. However, massive NGS has been introduced in diagnostic laboratories, thus revolutionizing the screening of these disorders. In addition, genetic diagnosis provides important information of the recurrence risk, allows the monitoring of disease progression, and gives the opportunity of timely treatment decisions, thus minimizing complications. Finally, the use of NGS to characterize microbiota may add further knowledge to a possible key factor that may be involved in the development and the progression of these diseases and open the way to novel therapeutic approaches.

The study of the microbiome, indeed, has provided further insights into the understanding of the immune mechanisms behind these diseases, thus representing a target for new therapeutic strategies. However, future studies are needed in order to deeply understand if a direct causal implication of the microbiome exists before the onset, or during, the early phases of autoimmune monogenic diseases. Several limitations also characterize the study of microbiota, since the mouse models that are mostly used in this field do not develop microbiota that are similar to humans. On the other hand, human individuals themselves show highly variable configurations of the microbiome, making the study of potential correlations between disease and microbe composition very challenging. This complexity opens the possibility of applying artificial intelligence and machine learning to characterize individual microbe patterns and thus create personalized therapies.

In addition to FMT, which has been widely used to treat *Clostridium difficilis* infections, new, more precise strategies are currently under investigation. For example, the elimination of specific pathobionts using bacteriophage therapy [175]; or the use of a personalized diet that alters the nutrient availability for specific strains, along with the intake of postbiotics, which represent the microbe-derived metabolites that are highly present in the systemic circulation, able to influence the immune system, as described in this review.

However, standardized and unbiased preclinical and clinical studies are needed before translating the microbiome-based treatments into clinical practice.

Lastly, the role of vitamin D in modulating the immune response and influencing microbiota composition suggests its potential use as a supplement within conventional therapies that are currently used to treat monogenic autoimmune disorders. Indeed, several studies have reported a reduced rate of autoimmune diseases, such as SLE, psoriasis, and rheumatoid arthritis, in patients under vitamin D treatment [176], thus encouraging the same approach in monogenic autoimmune disorders. However, studies in this field are currently missing and should be further investigated.

All together, these data highlight the importance of studying these rare diseases, which still leave numerous open questions.

## Figures and Tables

**Figure 1 biomedicines-11-01127-f001:**
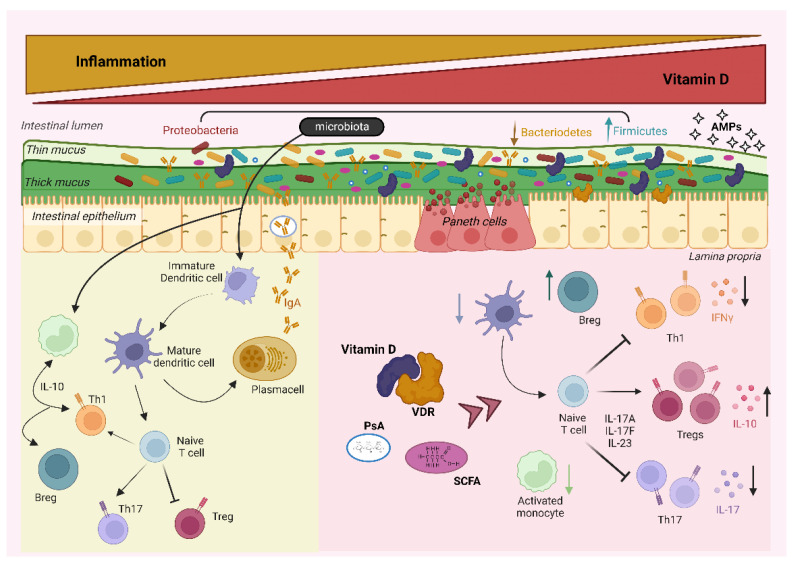
Gut microbiota and vitamin D modulation of the immune system. On the right side of the figure, an anti-inflammatory milieu, with a balanced microbiota composition, in the presence of normal vitamin D levels, which ensure intestinal barrier integrity, support a tolerogenic immune system setting. On the left side, a pathologic status with high levels of inflammation, low levels of vitamin D, and a decreased heterogeneity of microbiota, which promote immune cell activation while suppressing a regulatory phenotype. (AMPs, antimicrobial peptides; PsA, polysaccharide A; SCFA, short chain fatty acid; and VDR, vitamin D receptor). ↑ increased levels; ↓ decreased levels.

**Table 1 biomedicines-11-01127-t001:** Summary of monogenic autoimmune diseases.

Disease	Defective Gene(s)	Inheritance	Autoimmunity	Mechanism of Autoimmunity	Major Symptoms	Microbiota Changes
APS1 [42,49,50,61]	AIRE	Autosomal recessive	Organ specific	Defective Tregs Abnormal DCs Lymphocyte Infiltration AutoAbs	Hypoparathyroidism Mucocutaneous candidiasis Adrenal insufficiency	↓Firmicutes ↑Proteobacteria ↑Bacteroidetes
IPEX [69,72,80]	FOXP3	X-linked	Organ specific	Dysfunctional Tregs AutoAbs	Diarrhea Type I diabetes Thyroiditis Eczema	↓Firmicutes ↑Bacteroidetes
ALPS [92,97,98,110]	FAS FASL CASP10	Autosomal dominant Autosomal recessive (variable penetrance)	Systemic, organ specific	Defective lymphocyte apoptosis	Lymphadenopathy Splenomegaly Autoimmune Cytopenias Malignancy	N.A.
SLE [120,124,132,143,161]	Clq, Cls, Clr, C2, C4 TREX1 RNase H2 DNASE1L3 ACP5	Autosomal recessive	Systemic	Complement deficiencies Impaired apoptosis Nucleic acid degradation Nucleic acid sensing self-tolerance type I interferonopathies	Arthritis Cutaneous vasculitis Nephritis Infections	↓Firmicutes * ↓Lactobacillus * ↑Bacteroidetes * ↑Lachnospiraceae *

Abbreviations: ALPS, Autoimmune lymphoproliferative syndrome; APS1, Autoimmune polyendocrine syndrome type 1; AutoAbs, Autoantibodies; DCs, Dendritic cells; IPEX, Immunodysregulation polyendocrinopathy enteropathy X-linked syndrome; N.A., Not available; SLE, Systemic lupus erythematosus; Tregs, T regulatory cells; *, In polygenic SLE; ↑ increased levels; ↓ decreased levels.

## Data Availability

Not applicable.

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
