# Peer review of "Genes and Microbiota Interaction in Monogenic Autoimmune Disorders"

_biomedicines, 2023, doi:10.3390/biomedicines11041127_

Round 1

Reviewer 1 Report

The article is consistent within itself. The references are relevant and recent. The cited sources are referenced correctly. Appropriate and key studies are included. The paper is comprehensive, the flow is logical and the data is presented critically. Tables and figures increase the value of the paper.

However, there are some specific comments on weaknesses of the article and what could be improved:

Major points

1. The mechanism of autoimmunity in lines 30-42 is not a good and profound summary. There are plenty of other mechanisms.

2. The aim of the review - focus on the enlisted disease, but in what aspect? Microbiome alterations? Autoimmune features? Clinical presentation?

3. Inclusion of vitamin D in section 2 is not related and should be conceptualized more in details why you put this information there.

4.  Section 2 is too big and needs subsections to make the review more comprehensive.

5. Somewhere in section 2 you should add some connections to the next section - autoimmunity and autoimmune diseases

Minor points

1. The abstract is too plain and non-informative. Please, revise.

Author Response

Major points

  1. The mechanism of autoimmunity in lines 30-42 is not a good and profound summary. There are plenty of other mechanisms. We thank the reviewer for the comment. We added more details in the introduction about different autoimmune mechanisms.
  2. The aim of the review - focus on the enlisted disease, but in what aspect? Microbiome alterations? Autoimmune features? Clinical presentation? In this review we summarized the genetic features of monogenic autoimmune disorders and we explored literature data on the involvement of microbiota in the pathogenesis of these disorders, in order to highlight new possible therapeutic strategies and suggest open questions to be further investigated in this field.
  3. Inclusion of vitamin D in section 2 is not related and should be conceptualized more in details why you put this information there.  We appreciate the comment; in light of the well-known role of Vitamin D in the context of multifactorial autoimmunity and its influence on microbiota composition, we think it is interesting to report the available literature data, though limited, on Vitamin D in monogenic disorders as it could represent an interesting topic to be further explored in terms of new therapeutic strategies
  4. Section 2 is too big and needs subsections to make the review more comprehensive.  As suggested, we divided section 2 into two paragraphs in the revised manuscript
  5. Somewhere in section 2 you should add some connections to the next section - autoimmunity and autoimmune diseases.  We added a linking sentence at the end of section 2

Minor points

  1. The abstract is too plain and non-informative. Please, revise.  As kindly suggested, the abstract has been revised

Reviewer 2 Report

The authors present a broad review examining the relationship between the biota of the body and that of autoimmune disorders. Specifically, the authors hone in on the influence of microbial populations on an individuals immune system profile, and relate existing observations in such to those symptoms which are prevalent in the systems in which specific microbes reside.  There is a specific focus on how the balance of specific microbes may influence events though changes at the genetic level, with the potential of genetic screening touted as a means to determine susceptibility and risk of certain diseases to a persons microbial profile.

In reviewing this article, I made a number of observations. The authors should consider the following when preparing a suitable revision.

1.       There are times when I felt the writing could be better referenced. There are instances where there are large blocks of writing/paragraphs that are dependent on few references, and as such, I would have expecting referral to more instances that support the statements made. There are numerous instances of such, and together, the authors should review the entire manuscript and try to improve the degree of referencing throughout.

2.       Overall, the writing of the manuscript is good though there are instances where there are grammatical errors and some sentences are difficult to interpret. The authors should revise the language throughout the manuscript to address these issues.

3.       Some sections may benefit from having subsections added within them. At times, some sections branch out in different directions with a lot of content covered within. As such, the writing often goes ‘back and forward’ on some instances, and I believe implementation of structured subsections may improve the flow/structure of the piece overall.  

4.       The diagram is useful, however, the legend is light on details. This needs to be improved.

5.       The table needs references for the specific information provided.

6.       Some sections contain paragraphs that do not quite go into enough detail, or fit in with the text around them. In some instances these are short paragraphs, and in others they are single sentences. As mentioned above, structured subsections may address this issue, but if the authors opt to not follow this advice, a serious review of the flow of the manuscript is required.  

Author Response

  1. There are times when I felt the writing could be better referenced. There are instances where there are large blocks of writing/paragraphs that are dependent on few references, and as such, I would have expecting referral to more instances that support the statements made. There are numerous instances of such, and together, the authors should review the entire manuscript and try to improve the degree of referencing throughout. We appreciate the comment; we added references at the end of some sentences. However, when more than one sentence was referred to the same reference, we have added it only once, not to be redundant
  2. Overall, the writing of the manuscript is good though there are instances where there are grammatical errors and some sentences are difficult to interpret. The authors should revise the language throughout the manuscript to address these issues. We thank the reviewer for the suggestion, we revised the manuscript trying to simplify some sentences
  3. Some sections may benefit from having subsections added within them. At times, some sections branch out in different directions with a lot of content covered within. As such, the writing often goes ‘back and forward’ on some instances, and I believe implementation of structured subsections may improve the flow/structure of the piece overall.  We divided section 2 in different paragraphs, as for section 4. We hope the revised manuscript is now more fluent 
  4. The diagram is useful, however, the legend is light on details. This needs to be improved. In the revised manuscript we included a more explicative legend to the figure, as gently suggested by the reviewer
  5. The table needs references for the specific information provided. References have been added in the table of the revised manuscript
  6. Some sections contain paragraphs that do not quite go into enough detail, or fit in with the text around them. In some instances these are short paragraphs, and in others they are single sentences. As mentioned above, structured subsections may address this issue, but if the authors opt to not follow this advice, a serious review of the flow of the manuscript is required.   As mentioned in point 3, we divided section 2 and 4 in different paragraphs to make the reading more fluent 

Reviewer 3 Report

The current review discusses about systemic monogenic autoimmune disorders, including the autoimmune polyglandular syndrome (APS1), the immunedysregulation, polyendocrinopathy, enteropathy, X-linked syndrome (IPEX), the autoimmune lymphoproliferative syndrome (ALPS) and monogenic systemic lupus erythematosus (SLE), and also focuses on the involvement of microbiota in the pathophysiology of these disorders.

The review topic is relevant to the scientific community and also has the potential to advance the understanding of the subject matter. To our knowledge, there are currently no other published reviews on this topic, therefore this paper serves as a valuable contribution to the field. The review is written with great clarity, and the incorporation of a well-designed figure and table provides excellent aids to the readers. The structure of the paper is well thought-out, providing a logical flow of information for the reader.

 Minor point:

-As regards the pathogenesis of SLE, it would be useful to the reader to add a few more references, such as PMID: 36882218;  PMID: 36792346; PMID: 24493283, as they describe the preclinical and clinical involvement of type I interferons (and the TLRs involvement) in the pathogenesis of SLE.

Author Response

 Minor point:

As regards the pathogenesis of SLE, it would be useful to the reader to add a few more references, such as PMID: 36882218;  PMID: 36792346; PMID: 24493283, as they describe the preclinical and clinical involvement of type I interferons (and the TLRs involvement) in the pathogenesis of SLE. We thank the reviewer for the kind comments; as suggested, we added the references in section 4 to further support the involvement of interferons in SLE

Round 2

Reviewer 1 Report

The authors addressed all the comments and the paper has been improved significantly.

Reviewer 2 Report

The authors have responded suitably to my comments and the manuscript is closer to publishable standard.